# Facile Hydrothermal Synthesis of SnO_2_ Nanoflowers for Low-Concentration Formaldehyde Detection

**DOI:** 10.3390/nano12132133

**Published:** 2022-06-21

**Authors:** Chao Xiang, Tingting Chen, Yan Zhao, Jianhai Sun, Kaisheng Jiang, Yongzhen Li, Xiaofeng Zhu, Xinxiao Zhang, Ning Zhang, Ruihua Guo

**Affiliations:** 1State Key Laboratory of Nuclear Power Safety Monitoring Technology and Equipment, China Nuclear Power Engineering Co., Ltd., Shenzhen, Guangdong 518172, China; xiangchao@cgnpc.com.cn (C.X.); zhaoy@cgnpc.com.cn (Y.Z.); ningzhang@cgnpc.com.cn (N.Z.); 2State Key Laboratory of Transducer Technology, Aerospace Information Research Institute, Chinese Academy of Sciences, Beijing 100194, China; chentingting19@mails.ucas.ac.cn (T.C.); jiangkaisheng20@mails.ucas.ac.cn (K.J.); zhangxinxiao19@mails.ucas.ac.cn (X.Z.); 3School of Electronic, Electrical and Communication Engineering, University of Chinese Academy of Sciences, Beijing 100049, China; 4Institute of Urban Safety and Environmental Science, Beijing Academy of Science and Technology, Beijing 100054, China; liyongzhen@bmilp.com (Y.L.); zxf_402@163.com (X.Z.)

**Keywords:** hydrothermal, SnO_2_ nanoflowers, formaldehyde gas sensing, low concentration

## Abstract

In this work, SnO_2_ nanoflowers were prepared by a simple one-step hydrothermal process. The morphology and structure of SnO_2_ nanoflowers were characterized by SEM, TEM, Raman spectroscopy, and XRD, which demonstrated the good crystallinity of the SnO_2_ tetrahedron structure of the as-synthesized materials. In addition, the sensing properties of SnO_2_ nanoflowers were studied in detail. It was found that the SnO_2_ nanoflower-based gas sensor exhibits excellent gas response (9.2 to 120 ppm), fast response and recovery (2/15 s to 6 ppm), good linearity of correlation between response (*S*) vs. concentration (*C*) (lg*S* = 0.505 lg*C* − 0.147, R^2^ = 0.9863), superb repeatability, and selectivity at 300 °C. The outstanding performance can also be attributed to the high specific surface area ratio and size of SnO_2_ nanoflowers close to the thickness of the electron depletion layer that can provide abundant active sites, promote the rate of interaction, and make it easier for gas molecules to diffuse into the interior of the material. Therefore, SnO_2_ nanoflowers can be an ideal sensing material for real-time monitoring of low-concentration HCHO.

## 1. Introduction

Formaldehyde, one of the most potent carcinogens among indoor volatile organic compounds (VOCs), is a colorless, irritatingly odorous gas that is mostly sourced from furniture, building and decorating materials, and personal care products. Formaldehyde has been regarded as the most significant indoor air pollutant [1,2,3]. It has been shown that long-term living in an environment with excessive formaldehyde may lead to allergic dermatitis, immune system abnormalities, cancer, and other potential health issues [4]. The International Agency for Research on Cancer has classified formaldehyde as a Group I carcinogen, with a maximum permissible exposure limit of 1 ppm determined by the National Institute for Occupational Safety and Health (NIOSH) [5]. In the past few decades, various methods have been used for formaldehyde detection in conventional laboratories, such as chromatography, electrochemistry, colorimetry, fluorescence, and spectrophotometry [6,7,8]. However, most of these detection methods require high-precision testing instruments and must be conducted in a fixed testing environment, making it impossible to fulfill the need for real-time and on-site rapid monitoring of formaldehyde gas [8]. Therefore, it is of great significance to achieve good selectivity, reusability, reliability, and anti-interference on-site detection of low-concentration formaldehyde gas.

To date, formaldehyde gas sensors based on metal oxide semiconductors (e.g., SnO_2_, ZnO, In_2_O_3_, and NiO) have been increasingly favored in recent years for their high sensitivity, compact size, low cost, and mobility, making them ideal for real-time gas monitoring [9,10,11]. SnO_2_ is one of the most attractive n-type semiconductors with high- electron mobility, and superb stability, whose performance can be fine-tuned over a wide range by changing the crystal structure and morphology, dopants, contact geological attempts, and operating temperature or mode of operation [12,13]. SnO_2_ is generally a tetragonal rutile crystal structure at room temperature and a band gap of 3.5–4.0 eV that the crystal space group is *P*4_2_/*mnm* (No. 136) and the lattice constants are a = b = 4.737 Å and c = 3.186 Å, respectively [14,15]. Moreover, present formaldehyde gas sensors generally suffer from poor selectivity and long response and recovery time, making them unsuitable for real-time formaldehyde monitoring in the field. A unique nanostructure with a high surface area and porosity has been demonstrated to have a significant impact on the gas-sensitive performance of SnO_2_ nanoparticles [16]. As a result, nanostructured SnO_2_-based materials such as nanowires, nanorods, nanofibers, nanoflowers, and others have been extensively studied [17,18,19]. Among them, 2D or 3D nanostructures constructed from low-dimensional (0D, 1D, or 2D) materials are ideal for the detection of formaldehyde because of their higher specific surface area, which can provide more active sites. Guo et al. have demonstrated the synthesis and effective gas sensing properties of SnO_2_ microstructures assembled from porous nanosheets [20]. Li et al. prepared SnO_2_ nanosheets by aqueous solvothermal treatment, and then by ascorbic acid in situ reductions modified PdAu bimetallic nanoparticles doped to achieve highly sensitive detection of formaldehyde, and acetone gas for detecting formaldehyde and diabetes diagnosis [21]. Rao et al. performed selective microheater surface modification by fluorinated monolayer self-assembly to enable the controlled deposition of two-dimensional colloidal sphere arrays that were sacrificial templates for the chemical precipitation growth of metal oxides, and then used the microheater-induced thermal decomposition process to go out of the templates to synthesize ordered metal oxide hollow arrays, a sensor with a detection limit of 6.5 ppb and a response time of 1.8 s for formaldehyde [22]. Although there have been numerous reports on the synthesis of 3D SnO_2_ nanomaterials, most of the preparation processes are still very complicated and not conducive to commercial applications.

In this work, SnO_2_ nanoflowers were prepared by a simple and versatile hydrothermal method. The morphology and structure of the SnO_2_ nanoflowers were characterized in detail by scanning electron microscopy (SEM), transmission electron microscopy (TEM), Raman spectroscopy, and X-ray diffraction analysis (XRD). In addition, the gas-sensitive sensing performance of SnO_2_ nanoflowers was investigated using HCHO as the target gas. The novelty of our work is reflected in the following two aspects: facile preparation strategy and excellent gas-sensing performance. On the one hand, we used CTAB as a surfactant that can reduce the surface energy and thus change their relative stability to form new phases to mix SnCl_2_ and urea with a mixed solution composed of water and ethanol, and successfully synthesize SnO_2_ nanoflowers by moderating the reaction and annealing temperature [23]. It is well-known that sensors based on hierarchical porous structures generally have a large surface area ratio and a reduced tendency to form agaric platforms [24]. Therefore, the synthesis of porous SnO_2_ nanostructures with large surface areas and uniformly distributed crystal size is highly desirable for gas-sensitive materials. However, studies on the preparation of SnO_2_ with graded porous structures and the investigation of their sensing mechanisms/properties have not been widely reported. On the other hand, our sensor has good gas-sensitive performance, high sensitivity and reliability, short response and recovery time, and it can detect low-concentration of HCHO with a selectivity of >7 times concerning toluene (C_7_H_8_), ammonia (NH_3_), hydrogen (H_2_), and carbon monoxide (CO). In addition, the SnO_2_ nanoflowers synthesized by this method could have a high yield and low loss, and thus have an attractive potential for commercial applications. Finally, the gas-sensing mechanism of SnO_2_ nanoflowers was discussed.

## 2. Experimental Section

### 2.1. Regents and Materials

Stannous (II) chloride dihydrate (SnCl_2_·2H_2_O, ≥98.0%), Cetyltrimethylammonium bromide (CTAB, ≥99.0%), and urea (CH_4_N_2_O, ≥99.0%) were purchased from Sinopharm Chemical Reagent Co., Ltd., Shanghai, China. All of the chemicals were used, as received, without any further purification.

### 2.2. Synthesis of SnO_2_ Nanoflowers

SnO_2_ nanoflowers were prepared by a straightforward one-step hydrothermal method, in which we employed SnCl_2_ as the tin source, urea to provide a suitable alkaline environment, and CTAB as the surfactant. In a typical synthetic strategy, 2 mmol SnCl_2_·2H_2_O and 10 mmol urea were dissolved in a 40 mL mixture of water and ethanol (3:1) and stirred thoroughly to obtain a translucent colloidal solution. Subsequently, 2 mmol CTAB was added to the above solution and stirred slowly for 30 m to obtain a milky white solution. The colloidal solution was transferred to a 50 mL Teflon-lined stainless-steel autoclave and maintained at 160 °C for 20 h. After the precipitate natural cooling to room temperature, it was separated by centrifugation and washed several times with water and ethanol. Finally, the product was dried at 60 °C for 24 h and calcined at 400 °C for 4 h to remove the excess CTAB and further generate porous structures.

### 2.3. Characterization

The SU8020 microscope was used to perform the field-emission scanning electron microscopy (FE-SEM, Tokyo, Japan) measurements. An FEI Talos F200X with an accelerating voltage of 200 kV was used to acquire the transmission electron microscopy (TEM) images and high-resolution transmission electron microscopy (HRTEM, Waltham, MA, USA) images. The Raman spectroscopy was conducted in backscattering geometry with a Raman apparatus (Thermo Fischer XRD, Waltham, MA, USA) and the 514.5 nm line of an Ar^+^-ion laser as the excitation source. Cu Kα radiation (λ = 1.54056 Å, 40 mA, 40 kV, 6° min^−1^ from 10 to 80°) was used to create a powder X-ray diffraction (XRD) pattern by a Rigaku D/max2500 apparatus.

### 2.4. Gas Sensing Property Measurements

Firstly, the synthetic SnO_2_ nanoflower samples were mixed with a dispersant (deionized water) to form a paste. Afterward, the as-prepared paste was coated onto the forked finger electrodes of the micro-thermal plate to form a sensitive film with a thickness of about 3~5 um. The sensors were then sintered at 400 °C for 4 h and aged for 3 days under 3.0 V heating electricity to improve the stability and mechanical strength of the sensitive films. Finally, the operating temperature test of the gas sensor can be achieved by applying different heating voltages to both ends of the heating electrode; the gas sensitivity test of the sensor can be achieved by passing different standard gases into the test chamber. Furthermore, the sensor response value is defined as the ratio of the resistance of the sensor when placed in different gas environments, with the expression *S* = *R*_a_/*R*_g_, where *R*_a_ and *R*_g_ are the resistance of the sensor in air and the target gas, respectively. The response time and recovery time are defined as the time that it takes for the sensor to reach 90% of the total resistance change from its initial, respectively. The measuring circuit and test system for the MEMS gas sensor are shown in Appendix A. The test method of obtaining the desired gas concentration is described in detail in the Appendix A and previous articles [25].

## 3. Results and Discussion

### 3.1. Characterization

The morphological and structural changes of SnO_2_ nanoflowers before and after heat treatment were characterized by scanning electron microscopy (SEM). Figure 1a,b show SEM images before heat treatment, whereas Figure 1c,d show SEM pictures after heat treatment. From the low magnification SEM image (Figure 1a), it can be seen that the nanostructure of SnO_2_ is a flower-like structure made of nanosheets stacked in the presence of surfactant CTAB. As observed from the high magnification SEM image (Figure 1b), the surface of SnO_2_ nanosheets before heat treatment is smooth and the size is about 200–400 nm. After heat treatment, the nanostructure of SnO_2_ is unchanged as seen in Figure 1c, but the high magnification SEM image (Figure 1d) shows a large number of pores on the surface of the nanosheets composing the SnO_2_ nanoflowers with the size of 20–70 nm.

Figure 2 shows the HRTEM images of the SnO_2_ nanoflakes. The thickness of the SnO_2_ nanoflakes is about 5–8 nm, and the crystallographic planes of the nanoflakes are spaced at 0.33 nm and 0.26 nm, which distinguish the (110) and (101) planes corresponding to the tetragonal rutile SnO_2_ [16]. Moreover, the four diffraction rings of the SAED pattern shown in the inset correspond to the (110), (101), (211), and (112) planes, respectively. These could also confirm the tetragonal rutile structure of the SnO_2_ nanoflakes.

Figure 3a shows the Raman spectral images of the SnO_2_ nanoflower with Raman shifts at 323 cm^−1^, 1478 cm^−1^, 1630 cm^−1^, and 775 cm^−1^ corresponding to the E_u_, E_g_, A_1g_, and B_2g_ vibrational modes of SnO_2_, respectively [26]. The phase composition of the prepared samples was characterized by X-ray diffraction (XRD), as shown in Figure 3, and the XRD spectra of the SnO_2_ nanoflowers at Bragg cross (2θ) peaks of 26.61°, 33.89°, 51.78°, and 64.72° correspond to (110), (101), (211), and (112) tetragonal rutile SnO_2_ structures, respectively. It is following Joint Committee on Planar Powder Diffraction Standards (JCPDS) File Card No. 71-0652, which agrees with the SAED.

### 3.2. Gas Sensing Performance

The chemical activity of the semiconductor oxide sensor can be fully excited at the optimal operating temperature, so it strongly affects the response characteristics of the semiconductor oxide sensor. In order to find out the optimum operating temperature, the sensitivity of the SnO_2_ nanoflower sensor to 40 ppm formaldehyde gas is tested at different temperatures (160–360 °C), as shown in Figure 4a. We could draw the conclusion that the gas-sensitive sensor response increases to the peak as the operating temperature rises, and then shows a decreasing trend with the further temperature increase. Therefore, the optimal operating temperature of the SnO_2_ nanoflower sensor is 300 °C.

Figure 4b shows the response of the SnO_2_ nanoflowers gas sensor at several different HCHO gas concentrations from 0.25 ppm to 120 ppm at 300 °C. The response of the sensor shows a gradually increasing trend with the gas concentration increasing. The response of the SnO_2_ nanoflower sensor to 0.25 ppm formaldehyde is about 1.05, which indicates that the SnO_2_ nanoflower sensor is suitable for low-concentration formaldehyde detection. In addition, the fitted curve of the SnO_2_ nanoflowers response versus formaldehyde concentration is shown in Figure 4c. The sensor response increased almost linearly with the increase in formaldehyde gas concentration. The relationship between the response (*S* = *R_a_*/*R_g_*) and gas concentration (*C*) is fitted as: lg*S* = 0.505 lg*C* − 0.147, and the correlation coefficient R^2^ = 0.9863, where *C* is the concentration of formaldehyde.

The response and recovery time of the SnO_2_ nanoflower gas sensor are investigated for formaldehyde gas at a concentration of 6 ppm, as shown in Figure 4d. The response and recovery time are about 2 s and 15 s, which are much faster than most reported HCHO sensors [27,28]. This result may be attributed to the pore structure of the nanoflowers surface which can increase the specific surface area and thus facilitate the gas uptake/desorption and diffusion. Moreover, the reproducibility of the SnO_2_ nanoflower gas sensor is studied by testing the 140 ppm HCHO in three replicates under the same air conditions. The response/recovery time, as well as the response value (around 9.8), are almost reproducible as seen in Figure 4e. These results demonstrate that the sensor has excellent reversibility and reproducibility for HCHO.

Figure 4f shows the selectivity test of the gas sensor based on SnO_2_ nanoflower material at 100 ppm (300 °C) for interfering gases such as toluene (C_7_H_8_), ammonia (NH_3_), hydrogen (H_2_), and carbon monoxide (CO). We found that the sensor response to HCHO was 9.21, while the responses to C_7_H_8_, NH_3_, H_2_, and CO were 1.13, 1.23, 1.18, and 1.21, respectively. Furthermore, the response values to HCHO were 8.2, 7.5, 7.8, and 7.6 times higher than the response values to C_7_H_8_, NH_3_, H_2_, and CO, respectively. In conclusion, the SnO_2_ nanoflower-based sensor exhibited good selectivity for HCHO. In addition, we evaluated the best gas detecting performance based on SnO_2_ nanoflowers to that of other formaldehyde gas sensors described in recent literature (Table 1). Our sensor has high repeatability and a quick gas reaction time at 300 °C, according to the result.

### 3.3. Gas Sensing Mechanisms

The n-type metal oxide semiconductor SnO_2_ is one of the most representative gas- sensing materials, and its sensing mechanism can be explained by the chemical reaction of formaldehyde gas molecules on the SnO_2_ surface, which, in turn, causes a significant change in electrical conductivity [33,34,35]. In this work, the gas sensing mechanism of the SnO_2_ nanoflowers sensor is demonstrated in Figure 5, which was operated in non-vacuum conditions. When the sensor is exposed to air, some of the oxygen molecules are adsorbed on the surface of the gas-sensitive material, and they trap the free electrons in the SnO_2_ conduction band to form chemisorbed oxygen species (O_2_^−^, O^−^, and O^2−^). Furthermore, as the carrier (free electron) concentration decreases, an electron-depletion layer forms on the SnO_2_ surface, leading to a decrease in conductivity. The optimum operating temperature for SnO_2_ nanoflower is 300 °C, where O^−^ ions dominate, and the chemisorption oxygen ion reaction process can be expressed by Equations (1) and (2) [12]. When SnO_2_ nanoflowers are exposed to HCHO gas, the HCHO molecules will react with the adsorbed oxygen species (O^−^) (Equation 3) and re-release the previously trapped electrons into the conduction band of SnO_2_, which will result in a narrowing of the electron depletion layer, causing an increase in conductivity. Moreover, to illustrate the significance of surface oxygen, the response characteristics to formaldehyde gas at 60 ppm were tested by placing the sensors in air and N_2_ atmospheres, respectively, while at the same operating temperature and test environment (ambient temperature of 27 °C and relative humidity of 34%). It was found that the resistance value decreased by 42.3% and the response to formaldehyde gas dropped by 36.5%, while minimal change in response and recovery time when the sensor was placed in N_2_ compared with in air, as shown in Table 2. It can be analyzed that the lower oxygen concentration in the test environment leads to a decrease in the amount of oxygen adsorbed on the surface of the material, releasing some of the electrons into the conduction band of SnO_2_, which in turn macroscopically shows a decrease in the resistance of the material; at the same time, the decrease in the amount of oxygen adsorbed on the surface also leads to a reduction in sensitivity to the gas. These phenomena provide key evidence that surface adsorbed oxygen plays a crucial role in the gas-sensitive mechanism of the sensor.
O_2_ (gas) → O_2_ (ads),(1)
O_2_ (ads) + 2e^−^ → 2O^−^ (ads),(2)
HCHO + 2O^−^ (ads) → CO_2_ (gas) + H_2_O (gas) + 2e^−^(3)

Furthermore, in our work, the improved gas-sensitive performance of the SnO_2_ nanoflower sensor can be attributed to two aspects: (1) SnO_2_ nanoflowers are composed of many porous SnO_2_ nanosheets. On the one hand, SnO_2_ nanoflowers composed of porous nanosheets have a higher specific surface area than typical SnO_2_ nanoflowers, which can provide abundant active sites so that the lower detection limits. On the other hand, the porous structure makes it easier for gas molecules to diffuse into the interior of the material, which is conducive to improved response, and response times. (2) Suitable SnO_2_ nanosheet size. It has been shown that gas sensing performance can be significantly enhanced when the grain size of the sensing material is comparable to the thickness of the surface electron depletion layer (*L*_d_) [33]. According to the HRTEM image of the synthesized SnO_2_ nanoflake (shown in Figure 2), it can be seen that its grain size is about 8 nm, which is close to 2*L*_d_ (~ 3nm for SnO_2_ at 300 ℃) [22]. This allows the electron depletion layer to extend almost to the entire grain, thus promoting the rate of interaction between the HCHO molecule and the adsorbed oxygen ion and, improving the sensitivity of the sensor.

## 4. Conclusions

In summary, we have successfully prepared SnO_2_ nanoflowers in a large yield by a facile one-step hydrothermal strategy, where we utilized CTAB as a surfactant, and SnCl_2_ and urea were used as precursors. The morphology and structure of the SnO_2_ nanoflowers were characterized in detail by SEM, TEM, Raman spectroscopy, and XRD, which clarified the good crystallinity of the SnO_2_ tetragonal rutile structure. Because of the high specific surface area and the size of the SnO_2_ nanoflower close to the thickness of the electron-withdrawing layer, the SnO_2_ nanoflower-based gas sensor exhibits excellent gas response (9.2 to 120 ppm), fast response and recovery (2/15 s to 6 ppm), good linearity of correlation between response (*S*) vs. concentration (*C*) (lg*S* = 0.505 lg*C* − 0.147, R^2^ = 0.9863), superb repeatability, and selectivity at 300 °C. Our work shows that SnO_2_ nanoflowers are potential sensing materials for the real-time monitoring of HCHO.

## Figures and Tables

**Figure 1 nanomaterials-12-02133-f001:**
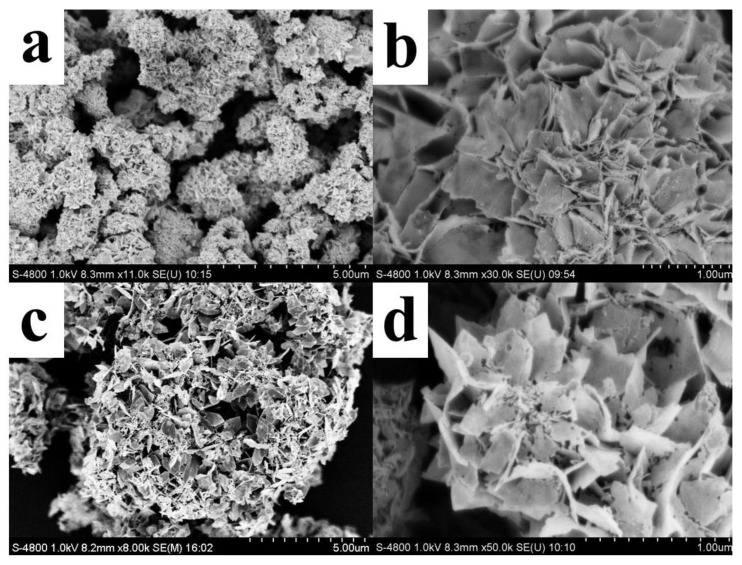
SEM images of SnO_2_ nanoflowers before and after heat treatment. Low magnification SEM images (**a**,**c**) and high magnification SEM images (**b**,**d**) before and after calcination at 400 °C.

**Figure 2 nanomaterials-12-02133-f002:**
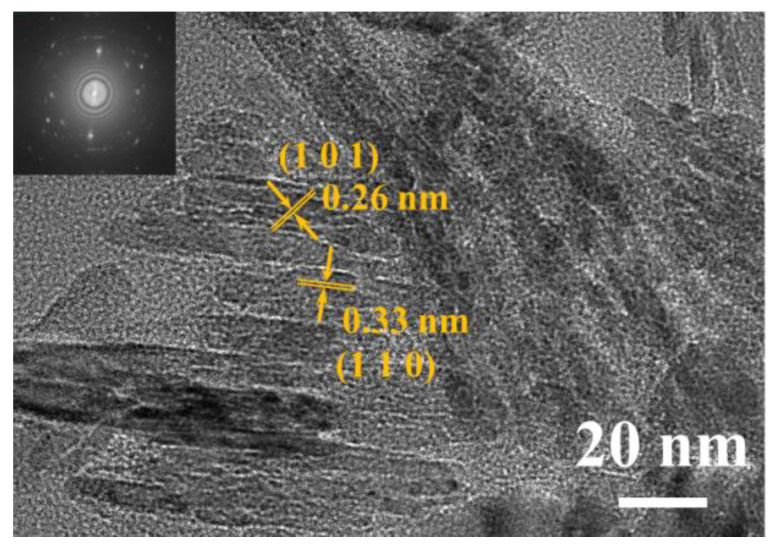
High-resolution HRTEM images of SnO_2_ nanoflowers. The inset shows the corresponding selected area electron diffraction (SAED) image.

**Figure 3 nanomaterials-12-02133-f003:**
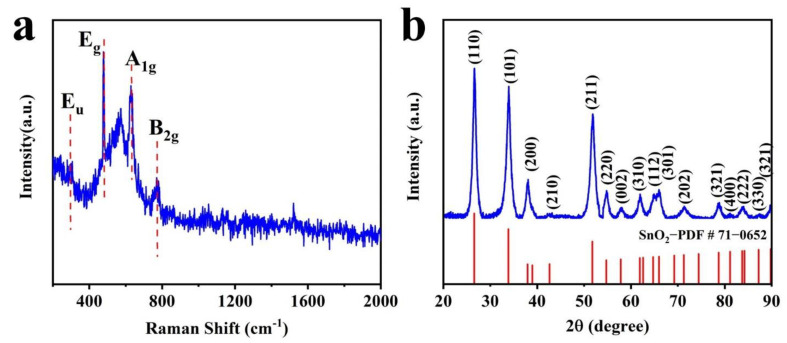
(**a**) Raman spectroscopy and (**b**) XRD images of SnO_2_ nanoflowers.

**Figure 4 nanomaterials-12-02133-f004:**
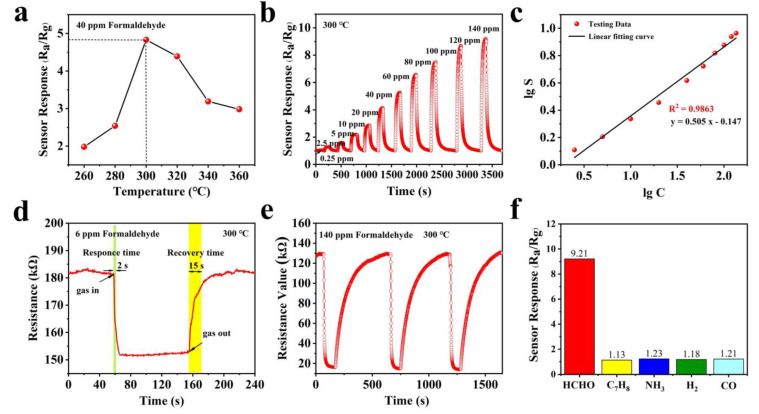
(**a**) Response of SnO_2_ nanoflowers sensor to 40 ppm formaldehyde at different temperatures. (**b**) Response of SnO_2_ nanoflowers sensor to formaldehyde gas at different concentrations. (**c**) The good linear plots of the SnO_2_ nanoflowers response versus formaldehyde concentration. (**d**) Response/recovery curve of SnO_2_ nanoflowers sensor for 6 ppm formaldehyde. The response and recovery time to 6 ppm HCHO are 2 and 15 s, respectively. (**e**) Three-time repeatability of SnO_2_ nanoflowers sensors. (**f**) The selectivity of SnO_2_ nanoflowers sensors for formaldehyde (HCHO), toluene (C_7_H_8_), ammonia (NH_3_), hydrogen (H_2_), and carbon monoxide (CO).

**Figure 5 nanomaterials-12-02133-f005:**
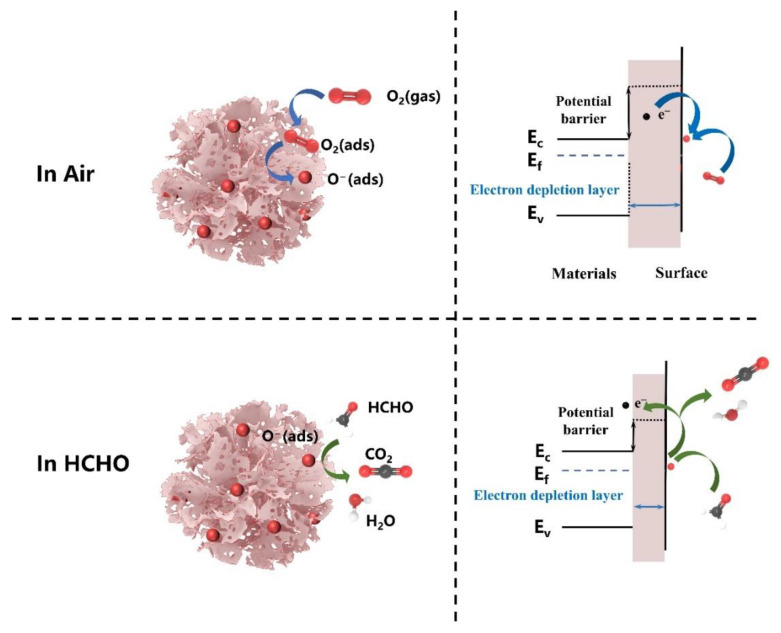
The gas-sensing mechanism of the SnO_2_ nanoflowers sensor. The left side is the schematic diagram, and the right indicates the energy band diagram.

**Table 1 nanomaterials-12-02133-t001:** HCHO sensing performance of different sensors.

Materials	Synthesis Route	Morphology	Temp. (°C)	Conc. (ppm)	Res.	Res./Rec. Time (s/s)	Ref.
PdAu/SnO_2_	hydro-solvothermal	3D nanosheets	110	100	125	68/32	[21]
SnO_2_	sacrificial template	hollow sphere array	300	0.5	~3	1.8/5.4	[22]
SnO_2_	hydrothermal	Cedar	200	100	13.3	<1/13	[23]
SnO_2_	solvothermal	mesoporous tubular	200	100	37	17/25	[29]
Pt/NiO	solution combustion	3D Porous	200	1000	8.2	102/70	[30]
MWCNTs-doped SnO_2_	sol-gel	nanometer-size powder	250	50	3.8	100/90	[31]
SnO_2_	hydrothermal	nanoparticles	230	50	35	20/23	[32]
SnO_2_	hydrothermal	nanoflowers	300	120	9.2	2/15	This work

**Table 2 nanomaterials-12-02133-t002:** The sensing performance of the nanoflowers sensor to 60 ppm HCHO in N_2_ and Air, respectively.

Test Condition	Response (R_a_/R_g_)	Resistance (kΩ)	Res. Time (s)	Rec. Time (s)
In N_2_	4.7	961.6	5.5	16.0
In Air	7.4	1667.1	5.0	18.0

## Data Availability

Not applicable.

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
