# Peer review of "Facile Hydrothermal Synthesis of SnO2 Nanoflowers for Low-Concentration Formaldehyde Detection"

_nanomaterials, 2022, doi:10.3390/nano12132133_

Round 1

Reviewer 1 Report

1-English needs moderate revision

2-References should be improved in order to show the innovations of the study

3-The authors should highlight the originality in the introduction 

4-The contributions of the investigation with respect to the state of art has to be included mainly in the conclusion and abstract

Author Response

Dear Reviewer 1,

We gratefully appreciate the editor’s and reviewers’ kindly suggestions to improve our work. We have studied the comments carefully and tried our best to improve the manuscript. Changes (marked in blue) have been made in the revision according to the reviewers’ comments. We sincerely hope our revision could meet the reviewers’ queries and are welcome further queries. Please feel free to contact us with any further questions. We are looking forward to your consideration. Please see the attachment about the point-by-point response to your comments. 

Thanks,

Tingting Chen

Reviewer 2 Report

The manuscript reported formaldehyde sensing with SnO2 nanoflowers. The results are straightforward without in-depth analysis. The novelty and scientific value are therefore limited. The authors should improve their manuscript by addressing the following issues:

1. In the introduction, the review on the previous sensing with SnO2 and other metal oxides should clarify what is the scientific issue. Why is SnO2 chosen for sensing? When SnO2 using for sensing, what about its phase, crystal structure, and microstructures. what is the sensing mechanism? which kind of issue is the research target?

2. "Although there have been numerous reports on the synthesis of 3D SnO2 nanomaterials, most of the preparation processes are still very complicated and not conducive to commercial applications". How complicated is it? Is this study for a commercial application? If true, what is the novelty of this research since hydrothermal synthesis is well known?

3. How is the sensor formed? Is the resistance of SnO2 nanoflowers bridging between the interdigital circuit measured? A schematic should be provided to illustrate the sensing method. how is the interdigital circuit connected with the meter? is the resistance measured by an ohm-meter with two-probe method or an I-V four-probe method?

 4. The sensing mechanism shown in Fig. 5 has to be evaluated. For example, if surface oxygen is critical, the sensitivity should be decreased in a vacuum. What is the purity of gases they used? 

Author Response

Dear Reviewer 2,

We gratefully appreciate the editor’s and reviewers’ kindly suggestions to improve our work. We have studied the comments carefully and tried our best to improve the manuscript. Changes (marked in blue) have been made in the revision according to the reviewers’ comments. We sincerely hope our revision could meet the reviewers’ queries and are welcome further queries. Please feel free to contact us with any further questions. We are looking forward to your consideration. Please see the attachment about the point-by-point response to your comments. 

Thanks,

Tingting Chen

Reviewer 3 Report

Referee Report 

on paper “Facile Hydrothermal Synthesis of SnO2 Nanoflowers for Lowconcentration Formaldehyde Detection

by authors

Chao Xiang, Tingting Chen, Jianhai Sun, Kaisheng Jiang, Yongzhen Li, Xiaofeng Zhu, Xinxiao Zhang, Yan Zhao, Ning Zhang and Ruihua Guo

submitted to Nanomaterials

This article makes a positive impression. Herein, SnO2 nanoflowers were prepared by a simple one-step hydrothermal process, their structure and sensing properties were investigated and comprehensively analyzed. With this work, the authors show that the SnO2 nanoflowers made can be an ideal sensing material for real-time monitoring of low-concentration HCHO. This work certainly deserves the attention of readers because of the high practical interest of this topic. In addition, the article is well constructed and easy to read. However, I have one major comment and a few minor ones that should be taken into account. After making corrections, the article may become suitable for publication ion Nanomaterials.

So, my decision is minor revision.

Comments

Major comment

1.     In this article, the outstanding performance exhibited can be attributed to the high specific surface area and size of SnO2 nanoflowers close to the thickness of the electron depletion layer. There is no doubt that the specific surface area is gigantic for nanomaterials. But this is not confirmed in the article. There are no direct measurements or indirect calculations. Because of this, it is not even possible to evaluate the prospects and advantages of one type of samples over others. I think this is a significant flaw that needs to be addressed. If the authors do not have the opportunity to make qualitative direct measurements, for example, by the gas adsorption method, then it is possible to make fairly accurate estimates of the specific surface area using SEM images. Several examples with detailed instructions can be found in DOI: 10.1016/j.jallcom.2021.158961 and DOI: 10.1039/d0ra05087c. Without specific surface area values, the conclusions seem speculative.

Minor comments

2.              All peaks on XRD pattern (Figure 3) should be identified.

3.              Please explain the motivation for choosing the SnO2 for nanoflowers fabrication. Apparently this is not the most promising material for gas sensory applications.

4.              The legend on the charts (Figure 3a, b and 4a,b,d,e) is redundant.

5.              The symbols in figure 3 should be enlarged or the resolution improved.

Author Response

Dear Reviewer 3,

We gratefully appreciate the editor’s and reviewers’ kindly suggestions to improve our work. We have studied the comments carefully and tried our best to improve the manuscript. Changes (marked in blue) have been made in the revision according to the reviewers’ comments. We sincerely hope our revision could meet the reviewers’ queries and are welcome further queries. Please feel free to contact us with any further questions. We are looking forward to your consideration. Please see the attachment about the point-by-point response to your comments. 

Thanks,

Tingting Chen

Reviewer 4 Report

This is a paper reporting the synthesis of SnO2 nanoflowers for formaldehyde detection. The paper could be of interest to readers. Nerveless some issues should be addressed before publication:

  1. The introduction section should be improved by highlighting the novelty in your research paper compare and linking your work with other previous research.

  1. Liping Gao et al Synthesis of SnO2 nanoparticles for formaldehyde detection with high sensitivity and good selectivity, Journal of Materials Research, 2020, DOI: 10.1557/jmr.2020.181
  2. Hai Yu et al, Facile synthesis cedar-like SnO2 hierarchical micro-nanostructures with improved formaldehyde gas sensing characteristics, Journal of Alloys and Compounds 724 (2017) 121-129

In addition, some more general information about the influence of the surfactant on the textural properties could be added. 

In the conclusion part, the authors state that “Due to the high specific surface area and the size of the SnO2 nanoflower close to the thickness of the electron-withdrawing layer, the prepared gas sensor exhibits excellent gas-sensing performance for low concentrations of HCHO at the optimum operating temperature (300 °C)”

Textural parameters should be presented if possible

Author Response

Dear Reviewer 4,

We gratefully appreciate the editor’s and reviewers’ kindly suggestions to improve our work. We have studied the comments carefully and tried our best to improve the manuscript. Changes (marked in blue) have been made in the revision according to the reviewers’ comments. We sincerely hope our revision could meet the reviewers’ queries and are welcome further queries. Please feel free to contact us with any further questions. We are looking forward to your consideration. Please see the attachment about the point-by-point response to your comments. 

Thanks,

Tingting Chen

Round 2

Reviewer 1 Report

I recommend the publication

Author Response

Dear Reviewer 1:

Thank you for your support to publish this manuscript "Facile Hydrothermal Synthesis of SnO2 Nanoflowers for Low-concentration Formaldehyde Detection" in Nanomaterials. On behalf of my co-authors, I would like to express our great appreciation for the time and effort that you dedicated to providing feedback on our manuscript. We are grateful for your insightful comments on and valuable improvement to our paper.

Yours sincerely,

Tingting Chen

Reviewer 2 Report

Although the authors have significantly improved their manuscript, they fail to provide the key evidence to explain the sensing mechanism. If surface absorbed oxygen species are critical for sensing they have to verify this by sensing in reduced oxygen atmosphere, i.e., the sensitivity will gradually decrease with decreasing oxygen. without this the manuscript does not display enough innovative. 

Reviewer 4 Report

The authors address satisfactorily all the issues raised. 

Author Response

Dear Reviewer 4:

Thank you for your support to publish this manuscript "Facile Hydrothermal Synthesis of SnO2 Nanoflowers for Low-concentration Formaldehyde Detection" in Nanomaterials. On behalf of my co-authors, I would like to express our great appreciation for the time and effort that you dedicated to providing feedback on our manuscript. We are grateful for your insightful comments on and valuable improvement to our paper.

Yours sincerely,

Tingting Chen

Round 3

Reviewer 2 Report

I do not see the proper manuscript and thus reject the current version. I would like to reminder the authors that the sensing mechanism has to be validated by your own experiments rather than by citated literature. If it is in the latter case, that means NO Novelty in your research at this point. The added experiment should be added in the main manuscript rather than in the supporting materials. 
